# Fine-Grained Cross-Modal Semantic Consistency in Natural Conservation Image Data from a Multi-Task Perspective

**DOI:** 10.3390/s24103130

**Published:** 2024-05-14

**Authors:** Rui Tao, Meng Zhu, Haiyan Cao, Honge Ren

**Affiliations:** 1College of Computer and Control Engineering, Northeast Forestry University, Harbin 150040, China; trlx20@nefu.edu.cn; 2College of Artificial Intelligence and Big Data, Hulunbuir University, Hulunbuir 021008, China; ske159@163.com; 3College of Information Engineering, Harbin University, Harbin 150076, China; zhum913@163.com; 4Heilongjiang Forestry Intelligent Equipment Engineering Research Center, Harbin 150040, China

**Keywords:** cross-modal, multi-task, image captioning, cross-modal retrieval, cross-modal alignment

## Abstract

Fine-grained representation is fundamental to species classification based on deep learning, and in this context, cross-modal contrastive learning is an effective method. The diversity of species coupled with the inherent contextual ambiguity of natural language poses a primary challenge in the cross-modal representation alignment of conservation area image data. Integrating cross-modal retrieval tasks with generation tasks contributes to cross-modal representation alignment based on contextual understanding. However, during the contrastive learning process, apart from learning the differences in the data itself, a pair of encoders inevitably learns the differences caused by encoder fluctuations. The latter leads to convergence shortcuts, resulting in poor representation quality and an inaccurate reflection of the similarity relationships between samples in the original dataset within the shared space of features. To achieve fine-grained cross-modal representation alignment, we first propose a residual attention network to enhance consistency during momentum updates in cross-modal encoders. Building upon this, we propose momentum encoding from a multi-task perspective as a bridge for cross-modal information, effectively improving cross-modal mutual information, representation quality, and optimizing the distribution of feature points within the cross-modal shared semantic space. By acquiring momentum encoding queues for cross-modal semantic understanding through multi-tasking, we align ambiguous natural language representations around the invariant image features of factual information, alleviating contextual ambiguity and enhancing model robustness. Experimental validation shows that our proposed multi-task perspective of cross-modal momentum encoders outperforms similar models on standardized image classification tasks and image–text cross-modal retrieval tasks on public datasets by up to 8% on the leaderboard, demonstrating the effectiveness of the proposed method. Qualitative experiments on our self-built conservation area image–text paired dataset show that our proposed method accurately performs cross-modal retrieval and generation tasks among 8142 species, proving its effectiveness on fine-grained cross-modal image–text conservation area image datasets.

## 1. Introduction

Neuro-networks function as parameterized databases, typically driven by specific tasks, with each network dedicated to fulfilling a corresponding task. However, there are instances where our requirements transcend single-task boundaries. Consider the context of rapidly accumulating natural conservation area image data. We seek not only to retrieve a single image but also to attach essential descriptions when summoning an image. Furthermore, we aspire to employ textual descriptions as queries to sift through our image repository, locating images that align with our specific needs. This scenario necessitates simultaneous engagement with two tasks: cross-modal image–text retrieval and image captioning.

As these data accumulate over time, the volume becomes formidable. For example, the Snapshot Serengeti Project at Serengeti National Park, Tanzania deployed hundreds of camera traps to understand the dynamics of African animal species. From 2010 to 2013, the project collected 3.2 million images from 225 camera traps [1]. And it was found to be very costly to manually process the images and add annotation labels, given such a large amount of data. The project carried out by Ref. [2] required thousands of technical volunteers to work for 2–3 months to annotate image data. With the improvement in camera manufacturing technology, each camera deployed in the field can record more than 40,000 photos per day due to a single trigger event [3], and many camera traps have been deployed in related projects. Refs. [4,5] deployed hundreds of camera traps in their project. Refs. [6,7] deployed about 50 cameras at water sources in natural conservation areas and recorded more than 800,000 wildlife images within a few weeks.

When we resort to two separate models to independently address these tasks, we encounter suboptimal outcomes. Specifically, the images retrieved through descriptive text queries may not align with the descriptive text generated by the model for the same image. In other words, these two models exhibit inconsistent encoding and decoding for the same data. Can we train a model that maintains consistency during both encoding and decoding, all while meeting the task requirements, thus mitigating semantic ambiguity within our cross-modal parameterized database?

To address this, we propose a multi-task model for joint training in cross-modal image–text retrieval and image captioning. Through the collaborative optimization of parameters, we achieve cross-module information sharing, thereby facilitating semantic-consistency encoding and decoding modeling. Post-training, the encoder and decoder can be independently employed to perform cross-modal image–text retrieval and image-captioning tasks while maintaining semantic consistency between the two tasks. This is made possible because our model is constructed upon a foundation of shared semantic-consistency representation space. Of course, the prerequisite is the construction of a dataset aligning with our specific needs and the judicious design of the model’s structure. For ease of exposition, we name the proposed method ReCap (Retrieval and Captioning).

As illustrated in Figure 1, we are able to retrieve corresponding images from the dataset using a customized textual input and subsequently generate descriptive text for the retrieved images. In this paper, our objective is to preserve semantic consistency in the context of fine-grained visual features and rich textual descriptions by jointly training a retriever and a captioner.

The contributions of this work include (1) the creation of a dataset of image–text pairs for natural conservation; (2) proposing a combined offline and online training approach; (3) introducing a method for information transfer through collaborative parameter solving within a multi-task module; and (4) presenting a technique for cross-modal alignment and semantic consistency preservation based on a shared representation space for cross-modal tasks.

## 2. Related Work

The cross-modal semantic consistency between images and text in our research is primarily achieved through the model design and joint training of two tasks: cross-modal retrieval and image captioning. The essence of this approach lies in the optimization of the cross-modal shared space embedding of images and text. On one hand, optimization is performed from the perspective of cross-modal alignment between image and text entities. On the other hand, the model needs to reorganize tokens related to the input image representation in the shared space in an autoregressive manner and output them in natural language, thereby achieving semantic consistency between image and text descriptions at a broader and deeper semantic level. The encoder and decoder constitute the core modules of our designed model, involving popular techniques in cross-modal alignment and cross-modal representation fusion. Subsequently, the literature review will delve into both cross-modal representation alignment and cross-modal representation fusion.

### 2.1. Cross-Modal Alignment

Currently, research on the cross-modal alignment of image and text representations is predominantly centered around contrastive learning methods. These studies achieve the embedding and alignment of image and text representations in a shared cross-modal space by training encoders separately for each modality using a contrastive learning loss. ConVIRT [8] demonstrates the potential of contrastive objectives to learn image representations from text. Inspired by ConVIRT, CLIP [9] performs pre-training on a dataset containing four billion image–text pairs and has become a milestone of vision–language models with excellent cross-modal representation. CLIP4Clip [10] demonstrates the CLIP model with high performance in cross-modal retrieval. ALIGN [11] performs pre-training on massive noisy web data. The above methods all use contrastive loss, which is the most effective loss for cross-modal alignment [12,13,14,15].

Intuitively, performing cross-modal contrastive learning by treating corresponding visual and textual entities as inputs to image and text encoders, respectively, can achieve better cross-modal alignment. Therefore, some research works in this domain utilize object detection models as visual unit extractors. The extracted target pixel regions are then fed to the image encoder for contrastive learning with the text encoder, enhancing the performance of cross-modal representations. Often, these studies require the integration of a pre-trained object detection model at the front end of the visual data input [16,17,18]. An intuitive approach is to align the visual features of the region where the object is located with the label. For example, Oscar [19] uses Faster R-CNN [20] to detect the object in the image and then aligns it with the word embeddings of the object tags. However, they are not suitable for fine-grained cross-modal alignment, as the object tags are too limited to align the vision features suitably. With a properly designed prompt, CLIP can be used for open-vocabulary classification, which solves the problem of limited object tags. ViLD [21] designed an open vocabulary object detection model by knowledge distillation from the CLIP. Ref. [22] achieved language-driven zero-shot semantic segmentation by directly using the representation of CLIP. Groupvit [23] implements unsupervised image segmentation by using the text representation of CLIP as a pseudo label.

Contrastive learning with dual encoders, while excelling in cross-modal retrieval tasks involving images and text, encounters challenges in adapting to fine-grained cross-modal retrieval tasks with natural conservation images due to the following reasons. First, certain species’ visual features in natural conservation images exhibit high intra-class and inter-class similarities, resulting in dense distributions of these highly similar representations in the shared space. This necessitates encoders with finer discriminative capabilities. Second, these encoders, trained on image–text pair datasets using contrastive learning, are often constrained by the representation of text descriptions alone and struggle to adapt well to cross-modal retrieval tasks where the semantics are similar but the expression methods differ.

### 2.2. Cross-Modal Fusion

With the successful application of the transformer [24] architecture in the fields of natural language processing, computer vision, and multi-modal, ViLT [25] proposes a transformer-based multi-modal encoder which focuses on cross-modal feature fusion, and takes the masked language modeling loss [26] for visual embedding as future work. This work has been achieved by VL-BEiT [27] after VIT [28] and MAE [29]. From then on, a big convergence of language, vision, and multi-modal pretraining has emerged. BLIP [30] proposes a new vision–language pre-training framework that transfers flexibly to both vision–language understanding and generation tasks. The multi-way transformer proposed by BEiT-V3 [31] has achieved state-of-the-art transfer performance in both vision and vision–language tasks. FLIP [32], which is called Fast Language–Image Pre-training, presents a simple and more efficient method for training CLIP by dropping a part of masked tokens. VLMo [33] jointly learns a dual encoder and a fusion encoder with a modular Transformer network. Coca [34] is a minimalist design to pre-train an image–text encoder–decoder foundation model jointly with contrastive loss and captioning loss like CLIP and SimVLM [35], respectively.

Cross-modal feature fusion is not suitable for cross-modal retrieval tasks due to the lack of effective optimization for unimodal encoders. However, when applied to image-captioning tasks for the same input image, this method generates descriptions that share the same semantics but have different expressions. This indicates that such methods contribute to solving cross-modal semantic consistency. Our research goal is to explore the joint application of cross-modal feature fusion and cross-modal feature alignment, aiming to leverage their respective strengths and compensate for weaknesses, fostering mutual enhancement. This objective is emphasized in the Method section for in-depth discussion.

## 3. Design Concept and Proposed Methodology

The overarching design strategy is to develop and train a pair of image–text encoders that extract representations with cross-modal semantic consistency, and the feature point distribution in the shared space accurately reflects contextual relevance. Based on this strategy, we designed a pair of encoders for cross-modal contrastive learning, consisting of an image encoder and a text encoder. After considering computational costs and performance trade-offs, we chose to obtain a pair of encoders through distillation that can be freely modified according to the experimental requirements (refer to the Section A.1 for detailed information). To promote cross-modal semantic consistency, we introduced the method of momentum encoding. However, the input data for cross-modal momentum encoding come from different modalities and lack mutual information, making it challenging to maintain consistency. To address this issue, we adopted a multi-task perspective and utilized a residual attention network to fully integrate representations from both modalities before outputting the momentum encoding queue. Finally, we trained the cross-modal encoder using a contrastive learning approach with the obtained momentum encoding queue to achieve fine-grained cross-modal semantic consistent representations. The overall architecture of the proposed method is illustrated in Figure 2.

Before introducing the cross-modal momentum encoder, we first present the residual attention network and the design of the objective function.

### 3.1. Residual Attention Neuro-Network

Based on Reference [36], we designed a residual attention network as illustrated in Figure 3. For detailed derivation of its input and output, please refer to Section A.2. The primary training objective for Cross-modal Res-Att is masked language modeling (MLM). In this context, let us denote a caption as CnP and the set of randomly masked positions as $M_{CnP}$. The MLM loss can be formally defined as follows:(1)$$L_{\mathrm{MLM}}=-\sum_{i\in M_{CnP}}\log p\left(CnP_{i}|CnP_{\setminus M_{CnP}}\right),$$
where $CnP_{\setminus M_{CnP}}$ is the masked version of the input caption, i.e., *Two* [MASK] *are* [MASK] *on* [MASK] *of a pond*. The cross-modal Res-Att module predicts the masked tokens based on the image and text context.

### 3.2. Captioner Training Objectives

The captioner is an autoregressive language generation model which operates on the principle of predicting the next token based on the input sequence and previously generated tokens. For example, given an initial input feature sequence $F^{k}=F_{1}^{k},\ldots F_{u}^{k}$ and the first token generated, denoted as $C_{1}^{k}$, the objective function is $\log p_{\theta }\left(C_{1}^{k}|F^{k}\right)$, and the generation process for $C_{2}^{k}$ is $\log p_{\theta }\left(C_{2}^{k}|F^{k},C_{1}^{k}\right)$, and so on. Therefore, the objective function of an autoregressive language generation model is represented by Equation (Equation 2):(2)$$L_{\mathrm{LM}}=\max_{\theta }\sum_{k=1}^{N}\log p_{\theta }\left(C_{1}^{k},\ldots \ldots ,C_{m}^{k}|F^{k}\right).$$
where θ represents the trainable parameters, and the input sequence $F^{k}$ can be visual features, language features, or a combination of both.

### 3.3. Image–Text Contrastive Loss Function

Following [8], the image–text contrastive learning (ITC) formulates the loss function according to InfoNCE [37]. Let *T* denote a certain species class embedding and *V* denote its visual embedding, then we have the embedding pair (V,T). We use $\left(V_{i},T_{i}\right)$ to denote the *i*-th pair of positive samples and $(V_{i},T_{j})j\neq i$ to denote a pair of negative samples. The ITC training objective of ReCap consists of two loss functions to make the distance of the positive pair closer than the negative one in the embedding space. Since ITC is asymmetric for each modality, it needs to be computed separately from both directions for images and text. The contrastive loss for the *i*-th pair in the image → text direction:(3)$$\ell_{i}^{\left(V\to T\right)}=-\log\frac{\exp\left(sim\left(V_{i},T_{i}\right)/\tau \right)}{\sum_{k=1}^{N}\exp\left(sim\left(V_{i},T_{k}\right)/\tau \right)},$$
where sim(·) is the cosine similarity, i.e., $sim\left(a,b\right)=a^{⊤}b/\left(∥a∥∥b∥\right)$, and τ is a temperature parameter. Similarly, we formulate the text → image loss as:(4)$$\ell_{i}^{\left(T\to V\right)}=-\log\frac{\exp\left(\mathrm{sim}\left(T_{i},V_{i}\right)/\tau \right)}{\sum_{k=1}^{N}\exp\left(\mathrm{sim}\left(T_{i},V_{k}/\right)\tau \right)}.$$

Finally, the training objective is a weighted sum:(5)$$L_{\mathrm{ITC}}=\frac{1}{N}\sum_{i=1}^{N}\left(\lambda \ell_{i}^{V\to T)}+\left(1-\lambda \right)\ell_{i}^{\left(T\to V\right)}\right),$$
where λ∈[0,1] is a hyperparameter weight, and N is the batch size.

### 3.4. Cross-Modal Momentum Encoder

In reference to the MoCo momentum encoding [12], we propose an offline encoder training method. Due to the high compression and ambiguity of textual information, compared to visual information, which is sparse, many detailed visual features are overwhelmed by dense textual information during cross-modal learning. To address this, we employ a residual attention network to repeatedly fuse visual features with textual features in a residual manner, increasing the proportion of visual information in the deep neuro-network’s forward channel. This enhances visual information redundancy to mitigate the drowning of sparse visual information during fusion with textual information. Additionally, because images contain factual information and exhibit invariance, aligning variable and ambiguous linguistic features around factual information contributes to eliminating linguistic feature ambiguity in context during cross-modal alignment. Consequently, this results in semantic consistency embedding, with visual information as the clustering center in the cross-modal representation space.

#### 3.4.1. Momentum Encoder

In brief, the principle of the momentum encoder is that the training of the encoder in unsupervised learning can be simplified as a look-up table problem. In other words, an encoded query should have high similarity to its corresponding key and low similarity to other keys. This simplifies the entire process to minimizing the contrastive loss. During the solving process, contrastive learning requires a queue containing keys for both positive and negative samples to look up for queries. To maintain the consistency of encoding for positive and negative samples in the queue, the momentum encoder is employed.

The encoding update rule for the momentum encoder is shown in Equation (Equation 6), where the momentum parameter m∈[0,1) is used. The query encoding $\theta_{q}$ is updated based on gradient back propagation, while the key encoding $\theta_{k}$ is updated using momentum. Typically, *m* takes a value greater than 0.9, which is equivalent to taking a moving average of the encoding updates. The slow-changing momentum encoder reduces the difference between the encoding of positive and negative samples in the queue, thereby improving the cross-task transfer performance of the encoder optimization process based on momentum in contrastive learning:(6)$$\theta \leftarrow m\theta_{k}+\left(1-m\right)\theta_{q}$$

#### 3.4.2. Offline Cross-Module Information Propagation

The cross-module joint solving of parameters constitutes the inter-module propagation of information. Deep learning models are essentially parameterized databases, with relationships among data implicitly encoded within the model’s parameters. Therefore, cross-module operations on parameters represent the propagation of information across modules.

Firstly, as illustrated in Figure 4, we feed the image–text paired dataset to the unimodal encoders, obtaining image encodings (Ve0,Ve1,Ve2…) through the ITC loss. Subsequently, as depicted in Figure 5, we feed (Ve0,Ve1,Ve2…) to the Res-Att module. Based on the momentum encoding method proposed in Section 3.4.1, a cross-modal momentum encoding queue is obtained using the joint loss function shown in Equation (Equation 7). Specifically, we obtain a visual momentum encoding queue (Vm0,Vm1,Vm2…) and a language momentum encoding queue (Tm0,Tm1,Tm2…). Then, as shown in Figure 6, we feed back the momentum encodings to update the unimodal encoders:(7)$$L_{\mathrm{Res}\&\mathrm{Cap}}=L_{\mathrm{MLM}}+L_{\mathrm{LM}}.$$

The loss function for the visual encoder in this training stage is represented by Equation (Equation 8), while the loss function for the language encoder is represented by Equation (Equation 9). The overall objective function is depicted by Equation (Equation 10):(8)$$\ell_{i}^{\left(V\to Tm\right)}=-\log\frac{\exp\left(sim\left(V_{i},{\mathrm{Tm}}_{i}\right)/\tau \right)}{\sum_{k=1}^{N}\exp\left(sim\left(V_{i},{\mathrm{Tm}}_{k}\right)/\tau \right)},$$
(9)$$\ell_{i}^{\left(T\to Vm\right)}=-\log\frac{\exp\left(\mathrm{sim}\left(T_{i},{\mathrm{Vm}}_{i}\right)/\tau \right)}{\sum_{k=1}^{N}\exp\left(\mathrm{sim}\left(T_{i},{\mathrm{Vm}}_{k}/\right)\tau \right)},$$
(10)$$L_{\mathrm{Momentum}}=\frac{1}{N}\sum_{i=1}^{N}\left(\lambda \ell_{i}^{V\to Tm)}+\left(1-\lambda \right)\ell_{i}^{\left(T\to Vm\right)}\right),$$
where λ∈[0,1] is the hyperparameter weight, and N is the batch size. Repeating these steps forms a closed loop for cross-modal momentum encoder training, which can be conducted offline. It should be noted that our proposed offline training method needs to be accompanied by the decoupling of the momentum encoding queue we adopted. This decoupling allows for independent settings of batch size and the length of the momentum encoding queue. For instance, during training, we used a batch size of 32 and a queue length of 4096. This enabled us to contrast more negative samples, facilitating the model to learn representations closer to the domain distribution. The length of the queue can be adjusted based on computational resources. In summary, the decoupling + offline strategy balances computational resources and model performance.

#### 3.4.3. Why Contrastive Learning and Momentum Encoding

The objective of contrastive learning is specifically to distinguish between positive and negative samples. If the encodings of positive and negative samples come from different encoders or different training stages of the same encoder, the model may learn more about the differences between the encoders rather than the differences between the data. To ensure a fair comparison between positive and negative samples and optimize the features extracted by the encoder, consistency in the encoding of positive and negative samples needs to be maintained in a long queue. For example, in our model training, the length of the momentum encoding queue is set to 4096. Essentially, contrastive learning treats each sample as a multi-class classification task, thereby enhancing the flexibility of embedding cross-modal contextual information in a shared space. However, due to the inherent diversity and ambiguity of natural language expressions, ambiguity is inevitable. This challenge is particularly pronounced in image–text paired datasets, where the same image can be interpreted from various perspectives, leading to significantly different language descriptions with varying semantics. Therefore, there are significant challenges to achieving cross-modal semantic consistency in representation. From a model structure perspective, cross-modal representation is determined by the encoder, and the compression of data information by the encoder inevitably leads to information loss. This requires a balance between encoding efficiency and encoder performance.

Image captioning is a standardized task for cross-modal understanding, where the model generates corresponding language descriptions based on input image representations. The task inherently involves calculating the similarity between image and text representations, necessitating a shared semantic space for image–text cross-modal semantic alignment, similar to cross-modal retrieval tasks. In other words, sharing a semantic space is a fundamental prerequisite for both cross-modal generation and cross-modal retrieval tasks. When the factual information and diversity/ambiguity of natural language descriptions in images are projected into a shared semantic space, the goal is to enhance the mutual information between the two modal representations. As the mutual information between modal representations increases in this space, the performance of cross-modal retrieval and cross-modal generation models based on this representation space improves. To optimize cross-modal representation and shared space embedding for the captioner’s cross-modal understanding, we propose a multi-task perspective involving the joint training of image-captioning and cross-modal retrieval tasks. This approach ensures primary consistency in the shared representation space between the two tasks, thus facilitating improved cross-modal mutual information. If the two tasks are trained separately, although they may project into the same-dimensional space, they contain different information without an information-sharing process between the modalities, thus failing to effectively reduce discrepancies between the modalities. To establish an information channel, we employ dual momentum encoders. However, directly comparing the image momentum encoder and the text momentum encoder through contrastive learning faces challenges in ensuring cross-modal semantic consistency due to the different data properties between the two modalities. To synchronously and consistently update the momentum encodings of both modalities across modes, we propose using a residual attention network as a channel for cross-modal information exchange. Considering the sparsity of image data and the abstract nature of language data, we ensure that sparse data contribute proportionately to the information during the deep network’s feedforward process by using image features as residuals. Through cross-modal information fusion and momentum encoding, we obtain momentum encoding queues with higher cross-modal mutual information, resulting in better performance of the image–text encoder in cross-modal semantic consistency.

## 4. Experiments

Our method is named ReCap (Retrieval and Captioning). In this section, we primarily validate the effectiveness of our proposed method on standardization tasks using public datasets. Specifically, these tasks encompass image captioning and image–text retrieval on the COCO dataset, classification tasks on the iNaturalist2018 dataset, as well as image-captioning and image–text retrieval tasks on the iNaturalist2018 dataset.

### 4.1. Dataset Settings

We utilized the Karpathy split [38] of the MSCOCO dataset [39], comprising 123,000 images, with each image accompanied by five sentences as annotations. The iNaturalist2018 dataset comprises 8142 distinct species, each serving as an individual image classification category. It encompasses a total of 437,513 training images and 24,426 validation images. As this dataset initially lacked caption annotations, we conducted a comprehensive annotation effort, providing five sentences of description for each image. Furthermore, we annotated both the common name and the Latin name for each species. The specific process of enhancing the INaturalist2018 dataset is detailed in Section A.3. In Table 1, we present some examples of our annotated data.

### 4.2. Implementation Details

We utilized eight NVIDIA 3090 24G GPUs for the image–text encoder contrastive learning training process, with a queue length set to 4096 and a momentum parameter of 0.995. We employed the AdamW optimizer with a decay weight set to 0.02. The learning rate was the warm-up set to 1 × $10^{-4}$ for the first 1000 iterations and decayed in a cosine function manner to 1 × $10^{-5}$ for the subsequent iterations. The total training duration for the model was approximately 127 h.

### 4.3. Evaluation Metrics

The image caption model employs four widely recognized evaluation metrics, namely, BLEU (Bilingual Evaluation Understudy) [40], METEOR (Metric for Evaluation of Translation with Explicit ORdering) [41], CIDEr (Consensus-based Image Description Evaluation) [42], and SPICE (Semantic Propositional Image-Captioning Evaluation) [43]. Among these, BLEU4 segments sentences into four-word chunks to gauge the descriptive accuracy of the model-generated captions. METEOR, building on the foundations of BLEU, addresses the issue of excessive word matching while emphasizing word recall and precision.

CIDEr, primarily applied in the domain of image description, employs TF-IDF (Term Frequency-Inverse Document Frequency) to weigh each sentence fragment. It encodes the frequency ($E_{r}$) of a fragment in the reference description and the frequency ($E_{c}$) in the generated description. Subsequently, it computes the similarity between $E_{r}$ and $E_{c}$ to generate an evaluation score for the model.

SPICE, on the other hand, is an evaluation metric based on scene graphs and semantic concepts. It assesses the extent to which the model-generated description aligns with the entities, attributes, and relationships present in the image.

The image classification task on iNaturalist has only one label for each picture, denoted as $g_{i}$. The result predicted by the model is denoted as $p_{i}$, and the error rate is
(11)$$e_{i}=\min_{i}d\left(g_{i},p_{i}\right),$$
where d(·) is
(12)$$d\left(x,y\right)=\left\{\begin{array}{cc} 0 & \mathrm{if}\;x=y\\ 1 & \mathrm{otherwise} \end{array}\right.,$$
and the total score is
(13)$$\mathrm{score}=\frac{1}{N}\sum_{i}e_{i}.$$

### 4.4. Experiment Project Selection

The core idea of our proposed method is briefly summarized as follows. Firstly, through the joint training of cross-modal retrieval and image-captioning tasks, we obtain a momentum-encoded queue with a contextual understanding of image–text pairs. This serves as an information bridge to train a cross-modal image encoder and a cross-modal text encoder using contrastive learning methods. This pair of encoders forms the basis for cross-modal fine-grained semantic consistency, as they determine the extraction and embedding of representations of various modal data into a shared cross-modal semantic space distribution. After training, our model yields an image encoder, a text encoder, and a captioner, which are the three key modules of ReCap. Due to the absence of a standardized task on a common dataset that can comprehensively evaluate our proposed method, we selected several standardized tasks on public datasets to individually test the performance of the three key modules of ReCap. Conducting experiments on standardized tasks on public datasets facilitates comparison with state-of-the-art (SOTA) methods on leaderboards, which, on the one hand, validates the effectiveness of the proposed method and, on the other hand, allows for a level measurement through comparison. Specifically, the experimental section validates the effectiveness of the captioner through the image-captioning task on the MSCOCO dataset as shown in Table 2. The effectiveness of the image encoder and text encoder’s cross-modal representations is verified through cross-modal retrieval tasks as shown in Table 3. The effectiveness of the image encoder is validated through the image classification task on the iNaturalist 2018 dataset as shown in Table 4. Additionally, Table 2, Table 3 and Table 4 in the experimental section reflect the proposed method’s multi-task perspective.

### 4.5. Evaluation on the MSCOCO Dataset

We trained models on the MSCOCO dataset to perform image captioning and image–text retrieval tasks in order to validate the effectiveness of the proposed method. Table 2 presents the performance comparison of ReCap with state-of-the-art models in the context of image captioning. Here, B4 denotes BLEU-4, C represents CIDEr, M stands for METEOR, and S corresponds to SPICE. Further details are provided in Section 4.3. Table 3 illustrates the performance comparison of ReCap in image–text retrieval tasks against high-level models. Here, I2T denotes image-to-text retrieval, while T2I represents text-to-image retrieval. R@1, R@5, and R@10 respectively indicate recall rates for the top 1, top 5, and top 10 retrieval recommendations. The experimental results demonstrate that ReCap outperforms several state-of-the-art models, thereby validating the efficacy of the proposed method.

Based on the comparative data in Table 2, it is evident that ReCap demonstrates improved performance compared to others. Taking the scores in the B4 column as an example, the ReCap score is increased by nearly seven points. This improvement can be attributed to two main enhancements: firstly, the incorporation of an open vocabulary, meaning there is no restriction on the number of categories; and secondly, the Res-Att network excels in the fusion of cross-modal features, effectively emulating the representation style of the dataset. This results in a higher overlap between the generated captions and the ground truth.

As shown in Table 3, in the retrieval task of image to text, the R@1 score exhibits an improvement of approximately 8 to 14 percentage points compared to others. In the text-to-image retrieval task, there is an improvement of approximately 1 to 12 percentage points compared to others. This indicates a significant effect of the proposed method in the cross-modal alignment of image and text features. The improvement in text-to-image retrieval performance is relatively challenging due to the high information compression in textual data and the sparse nature of image data. When calculating mutual information, the same textual representation often exhibits similarity to a larger number of image representations. For instance, different models of cars appearing in images with similar backgrounds would have high similarity. To effectively differentiate between the brand and model of cars in the image, a finer-grained cross-modal alignment is required for text-to-image retrieval. Therefore, adopting an open vocabulary approach during the training of the image encoder is essential, as it avoids the limitations to a finite set of categories and proves crucial in the cross-modal modeling tasks involving image and text.

### 4.6. Evaluation on the iNaturalist Dataset

In accordance with the introduction, the motivation behind this study is to address the need for the cross-modal processing of vast quantities of imagery data from natural conservation. In order to assess the cross-modal alignment of the model’s representations between images and text, we opted to employ the image classification task on the iNaturalist2018 dataset. This section’s experiments were conducted independently using the image encoder and text encoder. Notably, the image encoder was originally designed without a classification head. To achieve classification, we employed a method that involves comparing the representations output by the image encoder with the prompt encodings generated by the text encoder.

The format of the prompts used is ‘a photo of <category>’, where ‘category’ corresponds to the category names in the dataset. In other words, for as many categories as there are in the dataset, there are corresponding prompts. In essence, our image classification approach assigns an image to the category with the highest similarity to its image representation. Specific experimental results are presented in Table 4. The experimental outcomes demonstrate that ReCap outperforms several state-of-the-art models, thereby confirming the the proposed method’s cross-modal alignment capability between image features and textual representations for species.

As shown in Table 4, ReCap demonstrates a performance improvement of approximately 1 to 7 percentage points compared to others. This indicates that our proposed method, employing an open vocabulary approach, is capable of handling image classification tasks on the iNaturalist Dataset. The experimental results not only affirm the effectiveness of our method in cross-modal representation alignment but also validate the feasibility of applying this approach to open vocabulary image classification tasks.

### 4.7. Evaluation on the NACID Dataset

After verifying the effectiveness of the above, the model was trained on the NACID dataset Section A.3 and the two tasks of image captioning and image–text retrieval were evaluated. The model performance scores are shown in Table 5 and Table 6. Through all the experimental results, it can be seen that the model has the ability to perform image captioning and image–text retrieval on the enhanced INaturalist2018 image–text pair dataset, which verifies the effectiveness of the ReCap model proposed in this paper.

### 4.8. Qualitative Evaluation

Next, we conducted qualitative experiments on cross-modal retrieval and generation using the NaCID test set. Additionally, to validate the effectiveness of the proposed method on natural protected area image datasets, we selected three image datasets from natural protected areas for zero-shot experiments.

The top 5 results for text-to-image retrieval are illustrated in Figure 7. Both non-target images and target images contain relevant content related to grassland and the target species. From the perspective of our application, we seek relatively open-ended retrieval results. This approach allows the model to continuously improve through small-sample learning in real-world applications. If the model were confined to strict one-to-one retrieval, it would lack practical utility.

As shown in Table 7, the captions generated by the model align well with the content of the test images, and the species names are consistent with the Latin names used in the training set. This intuitively demonstrates the model’s learning capability in the domain of image–text cross-modal alignment. In the fourth prediction, the bear species (Ursus arctos horribilis) occurred 24 times in the training set, but there were no caption annotations for “cubs” in the training data prior to GPT-2 fine-tuning. This underscores the importance of pre-existing knowledge within NLP models for image-captioning tasks, as it can provide additional information that is subsequently expressed in the form of generated descriptions. In the context of our approach, aligning image representations cross-modally in the pre-trained NLP decoder representation space leverages the rich knowledge of the NLP decoder for a deeper understanding of the images.

We conducted zero-shot experiments using three datasets related to natural conservations; refer to Appendix A
Table A2. The experimental procedure was as follows: Firstly, we designed sentences resembling “A photo of <species>” based on the dataset content. Subsequently, we performed text-to-image retrieval with these sentences and provided the retrieved images to the captioner for generating descriptive text. The experimental results are presented in Table 8. The experimental results indicate that the species names on the retrieval side, the species within the images, and the species names on the generation side are all consistent. This observation underscores that the features extracted by the image encoder and text encoder are aligned, and the semantics of the encoder and decoder are in harmony, visually demonstrating the model’s capabilities in cross-modal alignment and semantic consistency between text and images. Examining the generated captions reveals the decoder’s capacity for systematic descriptions of foreground and background elements. This is a result of the combined influence of the model’s prior knowledge and fine-tuning.

### 4.9. Ablation Study

The results of the ablation experiments are presented in Table 9. In the table, the term “C+C” indicates a direct connection between the encoder and captioner, where the visual representations generated by the encoder are used as input for the captioner. “C+R+C” signifies the bridging of encoder and captioner through the Res-Att module.

From the experimental results in the “C+C” row, it can be observed that the I2T and T2I performance on both datasets is relatively consistent, maintaining an average level. In comparison to the performance of ReCap, there is a slight decrease in T2I, while I2T and image captioning exhibit more substantial performance degradation. This suggests that when the encoder and decoder operate independently, the model’s performance heavily relies on the knowledge inherited from pre-trained models and the training process. However, without a channel for information transfer between them, they cannot leverage distinct task perspectives from each other to enhance each other’s performance.

Looking at the experimental results in the “C+R+C” row, there is a noticeable improvement in the performance of image captioning compared to the “C+C” row. This indicates that after a finer-grained cross-modal alignment of image and text representations at the micro-level, it becomes more favorable for the captioner to generate descriptions for images. It is evident that the Res-Att module significantly contributes to the optimization of cross-modal representation alignment and the refinement of shared semantic space embedding for text and images.

ReCap and the “C+R+C” configuration only differ in the presence of a momentum feedback loop in their model structures. From the experimental results, it is evident that there are overall performance improvements in the model, particularly in the I2T and image-captioning tasks. This suggests that the feedback information on the decoding side significantly aids in enhancing the performance of the encoder, resulting in substantial gains in the cross-modal alignment of image and text representations.

The improvement in image-captioning performance further illustrates that, after optimizing the encoder’s performance, it is possible to further enhance the decoder’s performance. From the perspective of data propagation, the encoder is at the front end, and the captioner is at the back end. With the addition of momentum feedback and Res-Att-based cross-modal fusion, the two form a feedback loop for mutual optimization.

## 5. Conclusions

The image–text representation initially undergoes coarse alignment through the encoder, followed by fine-grained alignment by the decoding side consisting of Res-Att and the captioner. Subsequently, the encoder is momentum updated based on the decoding side information, forming feedback from the decoding side to the encoding side, enhancing the quality of both the encoder and caption generation. The essence of this process lies in the sharing of a semantic space, where the decoder imparts its understanding of embedding similarities and categorization to the encoder. These insights are propagated to the encoder’s network parameters through momentum-based backpropagation. Furthermore, contrastive learning on the encoding side plays a crucial role. As mentioned earlier, the classification in contrastive learning is open-ended, with as many categories as there are samples. Such a classification method has no upper limit on granularity, compelling the encoder to learn subtle distinctions among samples as much as possible. Achieving this solely from the encoding side would be information bottlenecked, and this is where feedback from the decoding side effectively bridges the information gap. Experimental results also confirm the contribution of prior knowledge in the decoder during this process. In summary, the feedback from the decoding side, the prior knowledge in the decoder, and momentum updates collectively enhance the quality of feature extraction in the encoder. All of this coalesces into a shared semantic space embedding for the encoder–decoder, where both entities possess a shared and aligned embedding space, embodying the essence of semantic consistency.

The performance of both cross-modal retrieval in image–text pairs and generative models fundamentally depends on the quality of shared space embeddings. The main contribution of our proposed method lies in the effective fusion of the advantages of both tasks in the cross-modal shared space embedding of images and text through thoughtful model design. This approach is particularly suitable for scenarios where there are strict alignment requirements between the objects in the image and the vocabulary in the text. Moreover, it demands that the model can further associate the input image representation with a more extensive and semantically rich textual description along a longer logical chain. Our proposed method is well suited for such scenarios.

## Figures and Tables

**Figure 1 sensors-24-03130-f001:**
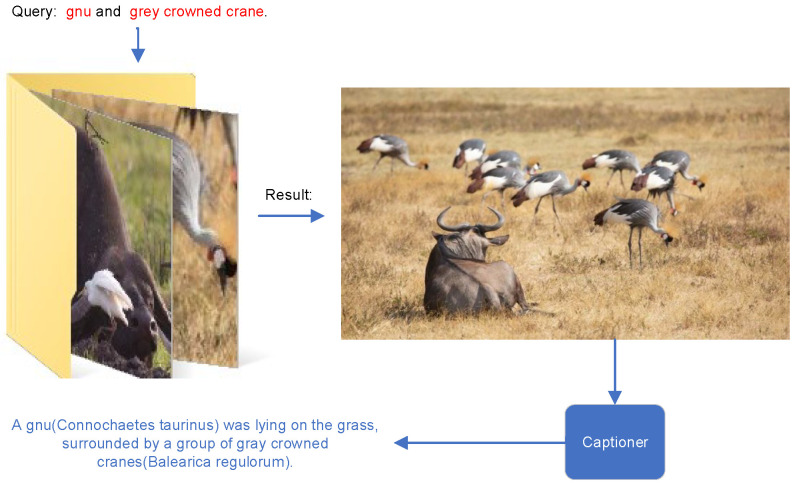
An application instance of the ReCap model.

**Figure 2 sensors-24-03130-f002:**
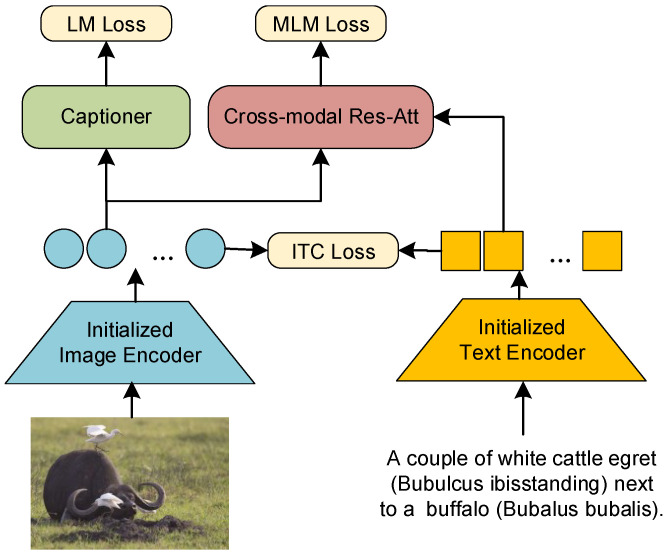
An overview of training ReCap for cross-modal semantics consistency.

**Figure 3 sensors-24-03130-f003:**
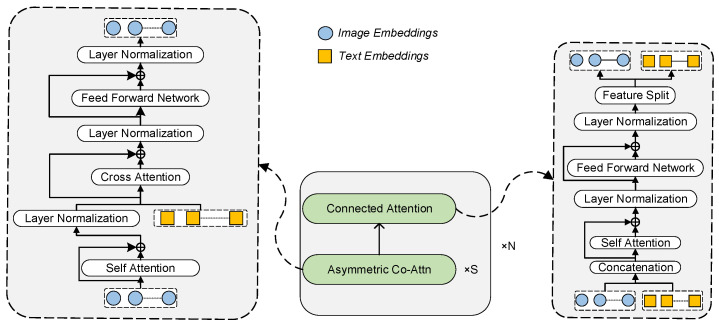
Residual Attention Network Architecture.

**Figure 4 sensors-24-03130-f004:**
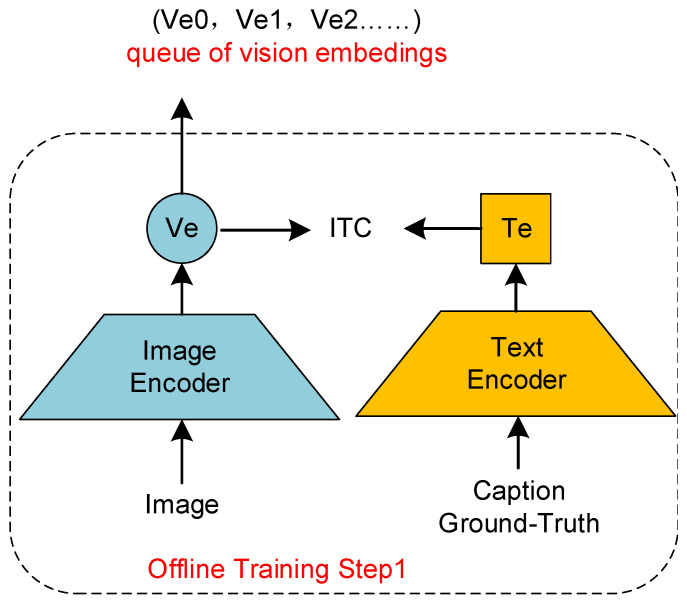
Initial visual encoding.

**Figure 5 sensors-24-03130-f005:**
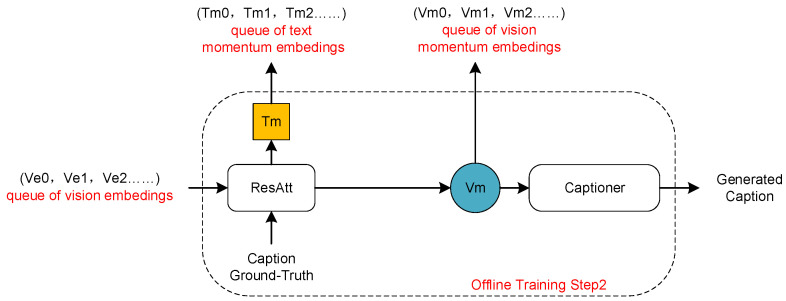
Redundant disambiguation momentum encoding.

**Figure 6 sensors-24-03130-f006:**
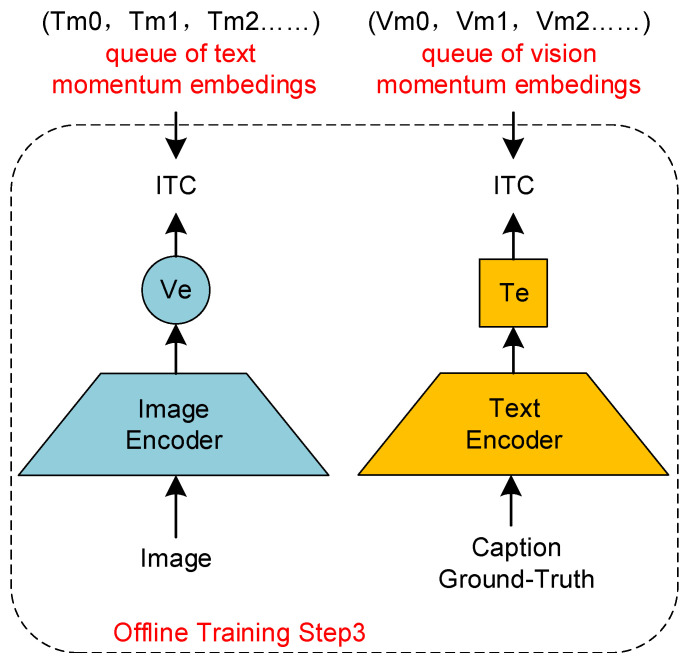
Unimodal encoding momentum update.

**Figure 7 sensors-24-03130-f007:**
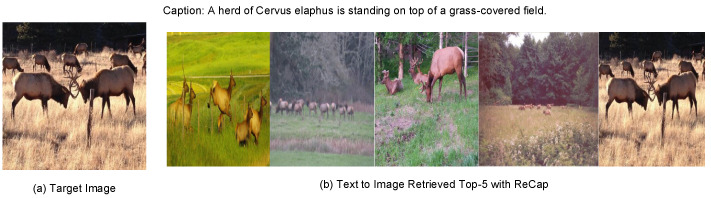
Examples of text-to-image retrieval on validation dataset.

**Table 1 sensors-24-03130-t001:** Samples of nature conservation image–text pair dataset.

Images	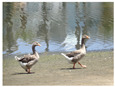	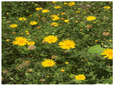	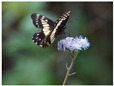	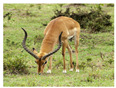
**Captions**	Two geese are walking on the shore of a pond.	A bunch of yellow flowers are sitting in a field.	A Catasticta nimbice is sitting on an Ageratum houstonianum in the sun.	An Aepyceros melampus grazing in a field.

**Table 2 sensors-24-03130-t002:** Quantitative analysis of image captioning on MSCOCO dataset (%).

Method	B4	C	M	S
Oscar [19]	36.6	124.1	30.4	23.2
BUTD [44]	36.2	113.5	27.0	20.3
UnifiedVLP [45]	33.53	113.1	27.5	21.1
ClipCap [46]	33.5	113.1	27.5	21.1
**ReCap**	39.8	126.7	31.6	24.4

**Table 3 sensors-24-03130-t003:** Quantitative analysis of cross-modal retrieval on MSCOCO dataset (%).

Method	Retrieval I2T			Retrieval T2I		
	**R@1**	**R@5**	**R@10**	**R@1**	**R@5**	**R@10**
Oscar [19]	57.5	82.8	89.8	73.5	92.2	96.0
METER [47]	57.1	82.7	90.1	76.2	93.2	96.8
ViSTA [48]	52.6	79.6	87.6	68.9	90.1	95.4
ALADIN [49]	51.3	79.2	87.5	64.9	88.6	94.5
**ReCap**	65.5	89.2	92.9	77.1	92.6	96.3

**Table 4 sensors-24-03130-t004:** Comparison on Image Classification on iNaturalist 2018 (%).

Method	Top1 Accuracy
MetaFormer [50]	84.3
OMNIVORE [51]	84.1
RegNet-8GF [52]	81.2
VL-LTR [53]	81.0
μ2Net+ [54]	81.0
MixMIM-L [55]	80.3
DeiT-B [56]	79.5
CeiT-s [57]	79.4
GPaCo [58]	78.1
**ReCap**	**85.1**

**Table 5 sensors-24-03130-t005:** Quantitative analysis of image captioning on NACID (%).

Method	B4	C	M	S
ReCap	40.8	144.1	33.6	25.5

**Table 6 sensors-24-03130-t006:** Quantitative analysis of cross-modal retrieval on NACID (%).

Method	Text to Image			Image to Text		
	**R@1**	**R@5**	**R@10**	**R@1**	**R@5**	**R@10**
ReCap	72.8	89.1	93.2	82.0	96.6	98.3

**Table 7 sensors-24-03130-t007:** Examples sentences generated by ReCap for test images.

Images	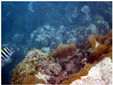	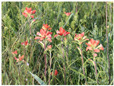	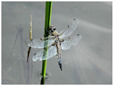	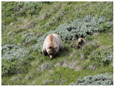
**Captions**	A few Abudefduf saxatilis swim in the stony water.	There are some red Castilleja indivisa in the grass.	A Libellula quadrimaculata is flying over the water.	A Ursus arctos horribilis and her cubs on a green field.

**Table 8 sensors-24-03130-t008:** Examples of ReCap zero-shot retrieval and captioning.

Query	A photo of Leopardus pardalis.	A photo of Phoenicopterus rubber.	A photo of Aglais io.
Dataset	Wildlife Conservation Society	Birds 510 Species-Image Classification	Animals Detection Images Dataset
Result	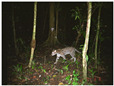	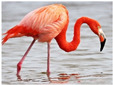	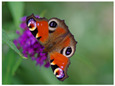
Caption	A small Leopardus pardalis walking through a forest at night.	A pink Phoenicopterus ruber standing in the water.	A close-up of an Aglais io is sitting on top of a flower.

**Table 9 sensors-24-03130-t009:** Ablation study of ReCap on the MSCOCO and iNaturalist 2018 datasets.

	MSCOCO			iNaturalist2018		
**Module Composition**	**I2T-R@1**	**T2I-R@1**	**Cap-B4**	**I2T-R@1**	**T2I-R@1**	**Cap-B4**
C+C	51.5	75.2	31.9	54.1	68.9	32.3
C+R+C	51.3	75.7	35.3	53.7	69.5	36.1
ReCap	65.5	77.1	39.8	63.6	72.2	41.0

## Data Availability

The raw data supporting the conclusions of this article will be made available by the authors, without undue reservation.

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
