# Peer review of "Fine-Grained Cross-Modal Semantic Consistency in Natural Conservation Image Data from a Multi-Task Perspective"

_sensors, 2024, doi:10.3390/s24103130_

Round 1

Reviewer 1 Report

Comments and Suggestions for Authors

Hi, this appears to be a solid piece of work. The authors describe the methodology well. I found the results to be highly compelling, in some cases though was the model overfitting? Did the authors take the necessary hyperparameter tuning methods to ensure this? Just looking for some further explanation of the results. Also, are there any computational performance indicators for the model? I would like the images to be displayed a little larger as well and with some improved resolution, if possible. With those changes I think it should be a good paper. 

Author Response

Response to Reviewer 1 Comments

  1. Summary

Thank you very much for your valuable time and selfless assistance. Your approval gives us more strength and confidence to continue our research journey. 

We will provide responses to your suggestions and comments in the following areas. Due to my lack of writing experience, the readability of the article was compromised. Therefore, the structural arrangement of the sections was adjusted, and the logical sequence of each subsection was clarified. Some content that affected readability was moved to the appendix. Key additions and modifications were highlighted in blue font.  The revised manuscript has been uploaded in PDF format to the Author's Notes File for your download and review.

2.Point-by-point response to Comments and Suggestions for Authors

Comments 1: [I found the results to be highly compelling, in some cases though was the model overfitting?]

Response 1: As you mentioned, we have always been concerned about overfitting during training. Specifically, here are some preliminary conclusions. First, to achieve granularity, we used a momentum encoder and employed the Res-Att module for cross-modal feature fusion when obtaining the cross-modal momentum encoding queue. Therefore, we consider this approach as a form of information leakage for standardized cross-modal tasks (such as image captioning tasks on the MSCOCO dataset and cross-modal retrieval tasks). However, our intention was to represent fine-grained cross-modal semantic consistency rather than solely achieving high scores. Secondly, we partitioned the MSCOCO and NaCID datasets in multiple ways, conducted cross-validation on multiple subsets, and monitored training and validation losses, without observing overfitting phenomena. We are particularly concerned about overfitting because we need a model that can be applied to real natural protected area image data; an overfitted model lacks generalization and thus loses its application value.

Comments 2: [Did the authors take the necessary hyperparameter tuning methods to ensure this?]

Response 2: Our approach bridges cross-modal information transfer using a cross-modal momentum encoding queue. During the acquisition of decoupled cross-modal momentum encoding queues, we jointly trained MLM and LM loss functions, employing a hyperparameter between 0 and 1 to adjust their proportion. Additionally, due to the nature of cross-modal momentum encoding, there is a need to merge and adjust the momentum hyperparameter m. A similar set of hyperparameters was also utilized when training the cross-modal encoder using contrastive learning loss functions. Grid search was employed for learning rate search. Other conventional hyperparameters such as decay and dropout are based on empirical observations and lack theoretical significance, thus not discussed in the paper.

Comments 3:  [Also, are there any computational performance indicators for the model?]

Response 3: The main computational performance indicators we consider include memory consumption and GPU resources. Thanks mainly to the decoupling of the momentum encoding queue, we can adopt an offline approach to train the entire model in a time- and step-wise manner. For instance, the representation of images can be pre-extracted, and the contrastive learning between the image encoder and the momentum encoding queue, as well as between the text encoder and the momentum encoding queue, can be conducted at different times and on different hosts. The advantage of this approach is that we can utilize multiple low-computational-power hosts with higher time flexibility to complete the entire experiment.

Comments 4:  [I would like the images to be displayed a little larger as well and with some improved resolution, if possible. With those changes I think it should be a good paper.]

Response 4: In the previous version of the manuscript, the figures were in PDF format. Based on your suggestion, we will export them from Visio in another format. Through further communication with the editor, we have decided on the format of the figure files, aiming to optimize the reading experience for readers to the greatest extent possible. Once again, we appreciate your valuable input for our research work.

Reviewer 2 Report

Comments and Suggestions for Authors

The authors claim to create a system similar to CLIP that provides solutions for text-and-image similarity queries, e.g., image classification, image recognition and image retrieval from text.

Contributions include:

1. creation of a dataset of image-text pairs for natural conservation

2. propose a combined offline and online training approach

3. introduce a method for information transfer through collaborative parameter solving within a multi-task module

4. present a technique for cross-modal alignment and semantic consistency preservation based on a shared representation space for cross-modal tasks.

For contribution (1) the authors may claim that they enhance the iNaturalist dataset. I feel it is not appropriate to claim that they have created a new dataset as the image data is derived from a pre-existing dataset and ground truth annotations have been added.

For contribution (2) it is unclear how the offline training approach is differentiated from competing similar approaches such as CLIP. For online training, this reader only sees the following reference "Of course, if computational resources permit, online processing is also feasible." A clear disposition of the novelty of online training needs to be included to be confident that there is novelty/contribution associated with some online training method.

For contribution (3), this reader found a description of a multi-task module in the methodology of this article but no evidence that the multi-task module has a contributory effect in terms of the provided results. An experiment might be required to demonstrate the impact that this approach has relative to an alternative solution. Perhaps this approach improves the accuracy shown in tables 2, 3 and 4 of results but it is not clear that this specific technique is the sole source of the improvement. Either a change of the contribution statement or a slightly different experiment highlighting this specific component would be better.

For contribution (4) this reader considers cross-modal alignment and semantic consistency preservation results from the results section as evidence. In this regard the results section has a small qualitative section specifically dedicated to the iNaturalist dataset which seems to be the focus according to the article title. While results are shown for COCO, the results for the iNaturalist dataset seem more important given the emphasis of the article.

Editorial Comments:

This reader felt that the contrastive encoder initialization section (3.1) was not clear about the purpose of using distilled encoders rather than a pre-trained text encoder. It would be appreciated if the authors can either provide some rationale in the methodology section or provide results in the ablation study.

Figure 2 created some confusion to the reviewer. From reading the paper, it is this reviewer’s understanding that the encoders for image-text matching and the captioning model combined are an integral system, where the output of the Res-Att network is fed to the captioner network. However, figure 2 appears to indicate that they are separate and are two individual systems.

On line 347, the authors say that they perform classification tasks, image captioning and image-text retrieval tasks on the iNaturalist 2018 dataset. However, only quantitative results of the classification results were provided in section 4.5. Also, it is not clear to the reviewer what dataset the data being evaluated in section 4.6 is from.

This reviewer felt that the flow of the article could be improved, especially in the methodology section. For example, perhaps all of the loss functions can be first discussed, i.e., moving section 3.4.1 to after section 3.3, then start another section to discuss details of the offline training.

What is Cap2Ret in the subcaption of Figure 9? 

Comments on the Quality of English Language

N/A

Author Response

Response to Reviewer 2 Comments

  1. Summary

We sincerely appreciate the valuable time and selfless assistance provided by the reviewer. Your insightful suggestions and meticulous comments have greatly aided us in reflecting on our experimental processes and optimizing the writing of the paper.

Below is our point-by-point response to the suggestions you provided. We will mark the corresponding modifications in blue font in the revised manuscript. To facilitate your review of the corresponding revisions in the revised manuscript, we have added a section after the 'Abstract' and before the 'Introduction,' titled 'Point-by-point response to Comments,' which provides hyperlinks for easy reference.  The revised manuscript has been uploaded in PDF format to the Author's Notes File for your download and review.

2.Point-by-point response to Comments and Suggestions for Authors

Comments 1: [For contribution (1) the authors may claim that they enhance the iNaturalist dataset. I feel it is not appropriate to claim that they have created a new dataset as the image data is derived from a pre-existing dataset and ground truth annotations have been added. ]

Response 1: We will change 'curating' the dataset to 'enhancing' it. Since the iNaturalist2018 dataset is already quite comprehensive, and we have only added caption annotations, we believe your suggestion is very apt. We are fortunate to have your oversight on our manuscript.

Comments 2: [For contribution (2) it is unclear how the offline training approach is differentiated from competing similar approaches such as CLIP. For online training, this reader only sees the following reference "Of course, if computational resources permit, online processing is also feasible." A clear disposition of the novelty of online training needs to be included to be confident that there is novelty/contribution associated with some online training method.]

Response 2: The premise of 'Offline' is to decouple the momentum encoding queue; if this cannot be achieved, training can only proceed online. Furthermore, after decoupling, batch size and the length of the momentum encoding queue can be independently set. For example, during our training, we used a batch size of 32 and a queue length of 4096. This allows for contrastive learning with a larger number of negative samples, facilitating the model to learn representations closer to the domain distribution. The length of the queue can be adjusted based on computational resources.

If computational resources allow, it is indeed possible to extract momentum encodings while simultaneously conducting contrastive learning." This is the concept we intend to convey in our manuscript. Clearly, as you pointed out, our expression was inaccurate and not conducive to reader comprehension. The corresponding modifications we have made will be highlighted in blue font in the revised manuscript.

Comments 3:  [For contribution (3), this reader found a description of a multi-task module in the methodology of this article but no evidence that the multi-task module has a contributory effect in terms of the provided results. An experiment might be required to demonstrate the impact that this approach has relative to an alternative solution. Perhaps this approach improves the accuracy shown in tables 2, 3 and 4 of results but it is not clear that this specific technique is the sole source of the improvement. Either a change of the contribution statement or a slightly different experiment highlighting this specific component would be better.]

Response 3: When solving cross-modal momentum encoding, we integrate image and text representations within the context formed by the combination of images and text in an image captioning task. Subsequently, we utilize a cross-modal retrieval task to train an image-text pair encoder using contrastive learning with the obtained momentum encoding queue.

After training the aforementioned model, we obtained an image encoder, a text encoder, and a captioner. In order to validate the effectiveness of these three key modules proposed in the method, the experimental phase involves the following validations:

(1). Validation of the captioner's effectiveness through an image captioning task on the MSCOCO dataset, as shown in Table 2.

(2). Validation of the effectiveness of the image encoder and text encoder's cross-modal representations through a cross-modal retrieval task, as shown in Table 3.

(3). Validation of the effectiveness of the image encoder through an image classification task on the iNaturalist 2018 dataset, as shown in Table 4.

Since there is no standardized task in any public dataset that directly aligns with the proposed method, we indirectly validated the method through standardized tasks on the aforementioned public datasets. This is because the essence of these tasks is to validate the performance of the encoders and captioner, and using standardized tasks from public datasets adds more credibility. Moreover, the experimental results in Tables 2, 3, and 4 also demonstrate the multi-task perspective of the proposed method.

Comments 4:  [For contribution (4) this reader considers cross-modal alignment and semantic consistency preservation results from the results section as evidence. In this regard the results section has a small qualitative section specifically dedicated to the iNaturalist dataset which seems to be the focus according to the article title. While results are shown for COCO, the results for the iNaturalist dataset seem more important given the emphasis of the article.]

Response 4: In addition, we supplemented experiments on the enhanced iNaturalist dataset. These additions will be highlighted in blue font in the revised manuscript.

Comments 5:  [This reader felt that the contrastive encoder initialization section (3.1) was not clear about the purpose of using distilled encoders rather than a pre-trained text encoder. It would be appreciated if the authors can either provide some rationale in the methodology section or provide results in the ablation study.]

Response 5: The adoption of distillation serves two purposes: firstly, at the time of our experimentation, CLIP was not open-source; secondly, we required an encoder that could be modified according to experimental needs after considering computational costs and performance trade-offs. The relevant discussions will be presented in the revised manuscript, highlighted in blue text.

Comments 6:  [Figure 2 created some confusion to the reviewer. From reading the paper, it is this reviewer’s understanding that the encoders for image-text matching and the captioning model combined are an integral system, where the output of the Res-Att network is fed to the captioner network. However, figure 2 appears to indicate that they are separate and are two individual systems.]

Response 6: The MLM Loss of Res-Att in Figure 2 and the LM Loss of the captioner are jointly solved, as shown in Equation 11 in the previous manuscript submitted. As you mentioned, our paper lacked clarity, which hindered readers' comprehension. You also noted this in your subsequent comments. To address this, we have reorganized the various sections of the paper, which will be reflected in the revised manuscript to be submitted later. Once again, we sincerely appreciate your valuable feedback, as it always proves beneficial to us.

Comments 7:  [On line 347, the authors say that they perform classification tasks, image captioning and image-text retrieval tasks on the iNaturalist 2018 dataset. However, only quantitative results of the classification results were provided in section 4.5. Also, it is not clear to the reviewer what dataset the data being evaluated in section 4.6 is from.]

Response 7: The classification task on the iNaturalist dataset is illustrated in Table 4 in the previous manuscript submitted. We have supplemented these experiments for captioning and image-text retrieval tasks in the revised manuscript, explicitly stating that they were conducted on the enhanced iNaturalist dataset, namely the NACID dataset.

Comments 8:  [This reviewer felt that the flow of the article could be improved, especially in the methodology section. For example, perhaps all of the loss functions can be first discussed, i.e., moving section 3.4.1 to after section 3.3, then start another section to discuss details of the offline training.]

Response 8: Your suggestion has been particularly insightful for us, as we lack writing experience and have tended to write only to the extent that we ourselves can understand, without considering from the reader's perspective how to present our arguments in a way that facilitates comprehension. Therefore, guided by your suggestion, we have reorganized the discussion sequence in each subsection to be more reader-friendly, aiming to enhance the fluency of reading and the ease of understanding. For some content that may hinder readability, we have moved it to the appendix and provided hyperlinks in the main text for reference.

Comments 9:  [What is Cap2Ret in the subcaption of Figure 9? ]

Response 9: To give a name to the model proposed by us, two options were considered: Cap2Ret and ReCap. This resulted in a typographical error, which we have corrected in the revised manuscript to uniformly use "ReCap".
